# The mediating effects of perceived social support and shame on psychological distress and its dimensions among Liberian refugees in Nigeria

Dogbahgen Alphonso Yarseah[1]*, Ololade Omolayo Ogunsanmi[2],
Joyce Olufunke Ogunsanmi[1], Olu Francis Ibimiluyi[1], Elijah Olawale Olaoye[3],
Esu Stanley Ezeani[4], Viola H. Cheeseman[5]

**1** Department of Guidance and Counseling, Ekiti State University, Ado Ekiti, Nigeria, **2** Department of Public Health, Babcock University, Abeokuta, Ogun, Nigeria, **3** Department of French, Ekiti State University, Ado Ekiti, Nigeria, **4** Department of HDSS, Disease Control and Elimination, The Medical Research Council Unit, The Gambia, **5** Department of Cyber Science and Engineering, Wuhan University, Wuhan, China

* dogbahgen.yarseah@eksu.edu.ng

**Data availability statement:** The dataset supporting the findings of this study has been made publicly available on GitHub at the following link: https://github.com/jamboyar/

## Abstract

Liberian refugees in Nigeria have faced prolonged displacement since the 1990s, experiencing significant psychological distress and shame. Perceived social support (PSS) is a crucial factor in mitigating these effects, yet limited research has explored its mediating role alongside shame in this population. This study examines how PSS and shame mediate psychological distress and explores their domain-specific interactions. A cross-sectional survey was conducted among 520 Liberian refugees (334 males, 186 females) in Oru, using structured questionnaires. Data were analyzed using SPSS and AMOS. Findings indicate that PSS partially mediates the relationship between shame and psychological distress ($\beta = -0.32$, $p < 0.001$), highlighting its protective role. Domain-specific interactions emerged, revealing that different dimensions of PSS and shame uniquely influence psychological distress. Family support ($\beta = -0.41$, $p < 0.001$) and friend support ($\beta = -0.28$, $p < 0.01$) were negatively associated with distress, while bodily shame ($\beta = 0.35$, $p < 0.001$) and characterological shame ($\beta = 0.47$, $p < 0.001$) were positively linked to distress. Notably, behavioral shame ($\beta = 0.21$, $p < 0.05$) was positively related to aspects of PSS, whereas family support showed no significant association with anxiety ($p = 0.12$) or bodily shame ($p = 0.08$). These findings emphasize the need for targeted mental health interventions accounting for the mediating role of PSS and the nuanced interactions between PSS, shame, and distress. The implications are discussed within the framework of Internal Family Systems (IFS) therapy, offering insights for refugee mental health support.

Mediating_effect_PSS The repository includes the full dataset in SPSS (.sav) format, along with a README file describing the variables and relevant supporting information. This is in accordance with PLOS's open data policy, ensuring transparency and accessibility for replication and further research.

**Funding:** The authors received no specific funding for this work.

**Competing interests:** The authors have declared that no competing interests exist.

## Introduction

Refugees worldwide face severe and persistent psychological challenges—often driven by unaddressed shame, lack of social support, and trauma-related stress stemming from forced displacement, loss, and prolonged insecurity. Among these populations, stateless Liberian refugees in Nigeria have endured three decades of displacement following Liberia's brutal civil wars, facing not only trauma during migration but also marginalization in host communities. Despite the formal closure of refugee camps, many Liberian refugees remain in legal and social limbo, with restricted access to healthcare, employment, and formal education. These enduring hardships contribute to long-term psychological distress, social isolation, and emotional vulnerability.

Psychological distress, comprising symptoms of depression, anxiety, and stress, is widely recognized as a key indicator of mental health burden among forcibly displaced populations [1,2]. Research by Yarseah [3] revealed that over 50% of Liberian refugees in Nigeria experienced high levels of emotional distress, particularly among women. Such distress is frequently rooted in traumatic experiences common among refugees, including witnessing death, losing family members, and exposure to sexual and physical violence [4,5]. These cumulative adversities not only undermine psychological resilience but also contribute to enduring emotional disruptions long after resettlement or relocation.

Among the emotional consequences of displacement, shame plays a critical yet underexplored role in refugee mental health. It is increasingly understood as a multidimensional construct, comprising characterological, behavioral, and bodily forms [6]. Characterological shame—the "shame of who I am"—involves a deep-seated sense of personal defect or worthlessness, especially pronounced among refugees who are often stigmatized or marginalized. Behavioral shame, or the "shame of what I have done," may stem from difficult choices made during conflict, such as abandoning loved ones or committing acts to survive, contributing to intrusive memories, guilt, and distress [7]. Bodily shame, the "shame of how I look," arises from visible signs of deprivation—malnutrition, illness, or poor hygiene—common in refugee camps, and may lead to social withdrawal and humiliation. These distinct dimensions of shame are not only psychologically burdensome but may also influence how individuals seek support and experience emotional distress.

In general, shame is a complex, self-conscious emotion involving internalized feelings of inadequacy, unworthiness, and self-blame [8]. These perceptions often derive from the sociocultural experiences of war and refugee life, which are associated with social stigma, dependence on humanitarian aid, unemployment, and a sense of lost dignity or autonomy [9,10]. Empirical studies consistently link shame to elevated psychological distress, making it a significant emotional burden among displaced populations [11,12].

Importantly, perceived social support (PSS) has been identified as a key buffer against the negative psychological outcomes of trauma and displacement. Support from family, friends, or community networks enhances emotional resilience and

reduces symptoms of depression, anxiety, and stress [13]. Among refugees, however, the quality and availability of perceived support can vary widely due to disrupted family structures, discrimination, and strained host-community relations. While previous studies often conceptualize PSS as a moderating factor—weakening the direct link between trauma and distress—such models may overlook its mediating role in helping individuals cope with complex emotions like shame. This study adopts a mediational framework, positing that perceived social support acts as a pathway through which shame influences psychological distress. That is, high levels of shame may diminish an individual's sense of being supported, thereby exacerbating emotional distress. All three constructs—shame, PSS, and psychological distress—are multidimensional, with distinct subcomponents that carry unique implications for research and intervention. Shame, for instance, may be categorized into behavioral, bodily, and characterological domains.

This study is guided by Internal Family Systems (IFS) theory [14], which conceptualizes psychological distress as arising from interactions among internal self-parts and emphasizes healing through self-compassion and integration. Perceived social support includes support from family, friends, and significant others, while psychological distress encompasses symptoms of anxiety, stress, and depression. Yet, few studies have examined the domain-specific components of these constructs within a unified model, especially in under-researched populations such as stateless Liberian refugees. In Nigeria, Liberian refugees continue to face economic marginalization, legal uncertainty, and social exclusion—all of which intensify experiences of shame and emotional distress. For these reasons, an in-depth examination of the emotional mechanisms underlying distress in this population is both timely and necessary.

## Research aim and contribution

This study investigates how multidimensional shame—specifically behavioral, bodily, and characterological shame influences psychological distress (anxiety, depression, and stress) through the mediating role of domain-specific perceived social support (from family, friends, and significant others) among stateless Liberian refugees in Nigeria. Guided by Internal Family Systems (IFS) theory, this research adopts a nuanced mediational framework to explore how internalized emotional burdens interact with external social resources to shape mental health outcomes in displaced populations.

By addressing a critical gap in refugee mental health research—namely, the domain-specific pathways linking shame to distress—this study advances theoretical understanding of emotional regulation and social buffering in the context of forced displacement. Practically, the findings offer culturally sensitive insights for designing interventions that target shame-based distress and enhance social support systems to promote psychosocial well-being among stateless refugee populations.

Therefore, the objectives of this study are:

1. To examine the mediating role of domain-specific perceived social support in the relationship between shame and psychological distress among Liberian refugees in Nigeria

2. To explore the multidimensional structure and interrelationships of shame, perceived social support, and psychological distress in this population.

## Based on these objectives, the following hypotheses are proposed

H1: Perceived social support (PSS) will mediate the relationship between shame and psychological distress among Liberian refugees in Nigeria.
H2: There will be significant associations between specific dimensions of shame (characterological, behavioral, bodily) and sources of perceived social support (family, friends, significant others).
H3: There will be significant associations between specific sources of perceived social support and dimensions of psychological distress (anxiety, depression, stress).

H4: There will be significant associations between dimensions of shame and dimensions of psychological distress.

## The mediating effects of perceived social support and shame on psychological distress

Psychological distress is a common mental health issue among refugees worldwide, with a high prevalence of post-traumatic stress disorder (PTSD) [4]. Psychological distress is a non-specific symptom of stress, anxiety, and depression [2]. A high level of psychological distress indicates impaired mental health and may reflect common mental disorders such as anxiety, depression, and stress [1]. Exposure to severe traumatic events is one of the leading causes of psychological distress among refugees. The loss of loved ones, destruction of property, insecure living conditions, war, torture, imprisonment, terrorist attacks, and physical and sexual abuse are among the traumatic experiences contributing to psychological distress in refugee populations [5]. For example, during the civil wars in Liberia, 89% of refugees in Nigeria witnessed a killing, 92% saw a family member die, and 45.2% of the females were raped [3].

Despite the growing body of literature on the buffering effects of perceived social support (PSS) on psychological distress among refugees [15–19], there is limited research investigating the underlying mechanisms of these relationships. PSS enhances refugees' mental health, self-worth, sense of security, and belonging [19]. It is a critical factor in refugees' sense of belonging and is often a primary influence in their decision to migrate to a foreign country [20]. This study assumes that PSS mediates the relationship between shame and psychological distress. PSS has been negatively associated with psychological distress among Nepalese refugees in Japan [17]. However, previous studies have not explored the interaction between PSS, psychological distress, and shame among refugees.

Shame has recently been included in the diagnostic criteria for PTSD in the DSM-V under the category of persistent negative "emotional states" [12] due to its avoidance and withdrawal tendencies. Shame is a painful, self-focused social-emotional experience linked to self-perception, social worth, self-identity, and one's position within a social group. It often stems from a fear of losing social status or failing to meet internalized standards [21]. During migration and post-migration processes, shame arises when personal characteristics such as nicknames, habits, and appearance conflict with one's psychological self-worth [21]. As a result, shame fosters an inward focus with deeply ingrained causes and promotes negative self-perception [21]. The emotional and distressing nature of shame has led to its strong association with psychological distress [8]. Additionally, shame overlaps with psychological distress and is considered a significant trigger of distress [9,10].

Given the strong relationship between shame and psychological distress, it is necessary to understand the mechanisms through which PSS influences this connection. Although shame has often been viewed negatively, scholars have also recognized its potential as a constructive emotion [21]. Healthy shame can foster positive attributes such as modesty, humility, gratitude, and respect for oneself and others. These characteristics help build meaningful and harmonious relationships, contributing to personal growth and transformation [21,22]. However, researchers have demonstrated that healthy shame can be easily distorted, leading to maladaptive emotional responses and toxic shame [23,24].

Despite these insights, there is a lack of research exploring the relationship between shame, PSS, and psychological distress among refugees. In the general population, studies on limited aspects of shame—such as stigma, bullying, and psychological distress—offer only partial evidence that shame directly affects perceived social support [25]. Notably, Kondrat et al. [26] found that PSS mediated the relationship between perceived stigma and mental health in the Canadian population. Furthermore, a recent study suggests that individuals experiencing shame are more likely to seek social connections [27]. This social approach and increased social awareness have led researchers to explore how shame may motivate positive interpersonal behavior as a means of restoring psychological self-worth [28,29].

Research has also indicated that shame can promote prosocial behaviors such as donating, cooperating, or giving gifts, particularly in situations where individuals feel ashamed in the presence of others [30,31]. Additionally, PSS has been found to regulate negative emotions [25] and enhance emotional well-being, which is crucial in alleviating psychological distress [26]. Psychological distress often coexists with shame [32], and high levels of PSS can mitigate the adverse

effects of stigmatization on mental health [33]. Consequently, strong PSS can help individuals affected by stigma develop positive self-awareness and reduce negative emotions [25]

## The dimensions of shame and their impact on Perceived Social Support (PSS)

Refugees are often exposed to pre-migration, migration, and post-migration shame, linked to extreme feelings of power-lessness, degradation, and humiliation [4]. These types of shame include beating, torture, hunger, sexual assault, loss of family members and friends, and traveling long distances—often considered traumatic shame []21. Shame is frequently experienced during wars in refugees' home countries and follows them through migration processes until they settle in the host nation [34]. Shame has various sources with divergent origins and requires different therapeutic management approaches [6,35]. However, research has often focused on the general aspects of shame [7] without considering these specific sources. Andrews et al. [6] suggested that shame can be categorized into three domains: behavioral, character-ological, and bodily shame [6,36].

As measured by Andrews and colleagues [6], behavioral shame arises from a failure in one's actions or inactions, such as begging a neighbor for food or money or engaging in socially dysfunctional behaviors. Koss and Figueredo [37] stated that behavioral shame involves feeling that one has acted inappropriately or transgressed against others and taking responsibility for the transgressionBehavioral shame often emerges among refugees due to experiences of sexual violence and other atrocities in the host nation, leading to degradation, embarrassment, or humiliation, and may also occur when refugees accept a devalued social identity or adapt to socially sanctioned values, beliefs, and norms [4,38]. Delezal and Gibson [21] argue that shame should be considered an umbrella term encompassing a range of emotions, including embarrassment, chagrin, mortification, and humiliation. On the other hand, behavioral shame can also serve a social func-tion by encouraging culturally acceptable values and behaviors [39]. Characterological shame is internal, stable, global, and self-deserving [40]. It is often referred to shame of who I am that involves an internalized sense of defectiveness, where individuals perceive themselves as inherently weak or incompetent, with attention directed inward [41]Character-ological shame may affect refugees' sense of self-worth and subsequently hinder not only their perception of available support mechanisms but also their willingness to seek social support. While research has shown that characterological shame predicts PTSD among Liberian refugees in Nigeria, the relationship between characterological shame and the various dimensions of perceived social support (PSS) remains unexplored in this population.

In contemporary society, physical appearance significantly influences romantic relationships and social opportunities [41,42], often leading to body surveillance and shame. Fredrickson and Roberts [43] define body shame as experiencing internalized negative views about one's body due to societal ideals of appearance. Bodily shame can lead to habitual body monitoring and self-surveillance, where individuals frequently assess their physical appearance [44]. Among refu-gees, bodily shame is often intensified by conditions such as starvation, malnutrition, amputations, tribal marks, lack of skilled labor, and chronic stress—factors contributing to PTSD as reported among Liberian refugees in Nigeria [45]. Bodily shame has been associated with psychological challenges such as body image dissatisfaction [46,47], eating disorders [47], and paranoid anxiety disorder [48]. Despite this, little is known about how bodily shame relates to perceived social support (PSS) within refugee populations. In the general population, however, PSS has been shown to positively influence body image, with broader support networks improving individuals' self-assessment of physical appearance [49].

Family support is often a key factor influencing refugee migration and plays a vital role in alleviating trauma-related symptoms [50,51]. Moreover, a high level of perceived family support has been shown to reduce bullying behavior among Canadian youths [52] Similarly, friendships become essential in the absence of family support. Friends provide emotional and informational support, offering a sense of identity and companionship during difficult times [53]. Adults, in particular, rely on friends for emotional support, practical guidance, companionship, and meaningful conversations during crises [54].

Additionally, significant others—including romantic partners, well-wishers, UN officers, and foreign nationals—play a crucial role in the support systems of refugees. Research indicates that support from significant others correlates with life

satisfaction among migrants [17]. Expressing emotions and receiving support from others enhances emotional well-being and contributes to happiness [55].

Although previous studies have not directly examined the relationship between shame and specific dimensions of perceived social support (PSS), our findings align with key tenets of Internal Family Systems (IFS) theory, developed by Schwartz [14,56]. IFS conceptualizes the mind as composed of multiple sub-personalities or "parts," governed by a core Self that embodies compassion, clarity, and healing. According to the theory, emotional distress—including shame—often arises when protective parts become polarized or extreme, disrupting the harmony of the internal system.

This framework provides a compelling lens through which to interpret our findings. Feelings of shame, as reflected in behavioral, bodily, and characterological dimensions, appear to interact with perceived deficits in social support, particularly from family, friends, and significant others. IFS suggests that when the Self is strong and integrated, individuals are better equipped to regulate shame and maintain supportive social connections. Thus, the mediating role of PSS observed in our study echoes IFS principles, particularly the capacity of a balanced internal system to foster healthier external relationships and mitigate distress [57].

Within IFS, psychopathology is seen as the activation of protective parts that serve survival functions to cope with distressing emotions and memories [58]. Behavioral expressions of shame can also promote culturally accepted values and social adaptation [39]. Thus, fostering self-compassion—linked to greater well-being and lower depression, anxiety, and stress [59,60] may serve as an effective strategy for improving social support and reducing shame.

Further, research suggests that individuals who can identify and harmonize with their internal parts demonstrate enhanced perspective-taking and prosocial behavior [61] supporting interpersonal connection and conflict resolution. Although a full exploration of IFS is beyond the scope of this paper, our study is the first to empirically connect dimensions of shame with dimensions of PSS through the lens of IFS principles. This integration highlights the potential for therapeutic approaches that resolve internal conflicts and strengthen social support networks in refugee populations.

## The dimensions of PSS on the dimensions of psychological distress

Psychological distress, which includes anxiety, depression, and stress, refers to mental health challenges arising from life-threatening experiences such as war, displacement, poverty, discrimination, and dehumanization [62]. Among Liberian refugees in Nigeria, research has documented particularly high levels of psychological distress; for instance, nearly half of male refugees (49.7%) and over half of female refugees (54.2%) report significant anxiety symptoms [3]. Anxiety itself is characterized by psychological, emotional, and cognitive-behavioral responses to perceived threats, often manifesting as restlessness, difficulty concentrating, and persistent anticipation of danger [63,64]. Moreover, anxiety has been shown to negatively correlate with perceived social support (PSS) among individuals facing life-threatening conditions [65]. Despite these findings, the relationship between PSS and psychological distress remains insufficiently explored in refugee populations.

Depression is characterized by low mood, sadness, helplessness, and pessimism [66]. People with depression often appear unkempt, feel inadequate, and experience psychomotor retardation [64]. Studies indicate that 61.1% of Liberian refugees in Nigeria suffer from depression. However, the literature is silent on the relationship between perceived social support from friends, family, or significant others and depression among Liberian refugees in Nigeria. Other studies in the general population, however, have shown that perceived social support from family is associated with lower levels of depression [67].

Furthermore, Cohen and Syme [68]observed that support from family, friends, and significant others serves as a protective factor that enhances well-being and aids recovery from illness. Specifically, social support provided by significant others to Syrian refugees in Germany has been associated with positive effects on mental health, whereas support from family reduces trauma reactivity [69].

Stress is a major mental health concern affecting approximately 51% of Liberian refugees in Nigeria [3](Yarseah, 2016). It refers to life changes or demands that exceed an individual's coping ability, leading to feelings of threat and vulnerability [70]. Refugees often face psychosocial and economic instability, unemployment, social discrimination, insecure residency, loss of family and friends, displacement from their communities, and challenges related to education and healthcare in host countries. These factors contribute to elevated stress levels among refugees (Ekmen et al., 2021) [15], which can lead to chronic stress. However, there remains limited understanding of how perceived social support (PSS) relates to stress among Liberian refugees in Nigeria.

PSS is role-specific, implying that individuals (family, friends, and significant others) are available to provide help in times of need [71]. Recent research on Syrian refugees in Turkey found that family support (a dimension of PSS) reduced stress [15], while support from friends predicted life satisfaction and, by extension, reduced anxiety and depression [15]. Similarly, Boge et al. [69] found that social support positively affects mental health and that support from family can mitigate traumatic experiences. Moreover, in the general population, emotional support has been shown to decrease physical distress, mental distress, depression, and anxiety [71]. Additionally, individuals who perceive themselves as receiving emotional support from close friends experience better mental health than those who do not [72].

## The dimensions of shame and their impact on psychological distress

The relationship between shame and psychological distress is significant. Shame is closely linked to various psychological symptoms, including anxiety, depression, loss of self-confidence, low self-esteem, anger, frustration, feelings of helplessness, and social withdrawal [10,73]. Studies have shown that symptoms of shame often overlap with symptoms of psychological distress, such as anxiety and depression [74]. Research indicates that Liberian refugees in Nigeria experience high levels of psychological distress, with approximately 76.4% suffering from PTSD [3]. PTSD is known to be comorbid with depression, anxiety, and stress. However, it remains unclear whether different subtypes of shame are specifically associated with various dimensions of psychological distress.

Behavioral shame refers to the negative evaluation of one's actions, which is specific, amendable, and closely linked to guilt [75]. As a socially and emotionally distressing experience, behavioral shame is often associated with past trauma and can trigger symptoms such as intrusive thoughts, flashbacks, emotional avoidance, hyperarousal, dissociation, and fragmented states of mind, and research on refugees has established a strong association between behavioral shame and PTSD [38]. Wilson, Drozdek, and Turkovic [76] argue that refugees often develop shame from past traumatic experiences, leading to feelings of fear, helplessness, and moral distress. Despite these negative effects, behavioral shame can also serve as a means of reinforcing culturally acceptable values, helping individuals align with social norms and expectations [39].

Previous research has shown that characterological shame is a predictor of PTSD among Liberian refugees in Nigeria [45]. However, its direct relationship with anxiety, depression, and stress among refugees has yet to be fully examined. Characterological shame, which is internalized and self-focused, has been consistently linked to stress, anxiety, and depression in the general population [77]. It is also associated with paranoia, other mental health disorders, and anxiety [48]. Additionally, characterological shame has been linked to emotional suppression, suggesting that individuals who experience shame related to their identity may develop maladaptive coping strategies to manage overwhelming emotions [78]. These maladaptive behaviors may, in turn, reinforce characterological shame and contribute to psychological distress [79]. Research further indicates that characterological shame is associated with low self-esteem, heightened self-criticism, verbal aggression, depression, and increased psychological distress [78,80,81].

There is a lack of research on the relationship between body shame and psychological distress—specifically depression, stress, and anxiety—among refugee populations. In the general population, body image dissatisfaction has received less attention among Africans compared to their European and Western counterparts [82]. However, studies suggest that body dissatisfaction contributes to psychological distress among African migrants in Europe [83] and is associated with PTSD among Liberian refugees in Nigeria [45].

Body shame involves social comparison, where individuals evaluate their bodies against others and experience feelings of inadequacy [84]. However, among refugees, body shame is not merely a cognitive or emotional experience but is often linked to physical deformities resulting from starvation, malnutrition, amputation, or torture. The consequences of body shame can be severe. For example, research has linked body shame to hostility, depression, psychological distress, and low self-esteem [78,81]. Additionally, it has been associated with difficulties in asserting sexual boundaries [78]. However, some studies have found no significant relationship between body shame, behavioral shame, and anxiety [19,48,84,85].

Despite the growing body of literature on shame and psychological distress, there is a notable gap in research exploring the specific relationships between different subtypes of shame and dimensions of psychological distress among Liberian refugees in Nigeria. Further studies are needed to examine how behavioral, characterological, and body shame contribute to anxiety, depression, and stress within this population.

## Materials and methods

### Ethical consideration

After obtaining ethical approval for the study, the Research Committee of Ekiti State University issued an official letter to the refugee camp authorities requesting permission to conduct the research. Upon approval, the camp authorities held a meeting with various camp leaders to schedule and facilitate data collection.Despite the general consent granted by the camp authorities, each participant was provided with an individual research consent form to read, ensuring confidentiality. Only those who voluntarily agreed to participate were included in the study.

As per the camp's regulations on research activities, each participant received two cups of rice, instant noodles, and a pencil as a token of appreciation. The estimated time for completing the questionnaire was 25–30 minutes. For participants who could not read or write, verbal assistance was provided by trained research confederates—university graduates fluent in Liberian dialects. Five research confederates were engaged in this study, each receiving a stipend of $10 per day. To minimize social desirability bias and interviewer influence, trained research assistants followed standardized, non-directive scripts and provided verbal assistance only when necessary. Participants were assured of confidentiality and encouraged to respond honestly, with privacy maintained during responses

Due to the sensitive nature of the questionnaire items, a counseling session was made available for three weeks during data collection to support participants experiencing emotional distress as a result of their participation. Additionally, special financial assistance was provided to individuals in critical conditions, regardless of whether they participated in the study.

### Trauma sensitivity consideration

Given the sensitive nature of the topics explored, including psychological distress and shame, the data collection process was designed with trauma sensitivity at its core. Research assistants provided verbal support following trauma-informed, non-directive scripts to ensure interactions were respectful, empathetic, and non-judgmental. Participants were clearly informed that they could skip any question or withdraw from the study at any time without consequences. Privacy and confidentiality were strictly maintained throughout to create a safe and supportive environment conducive to honest responses. Additionally, counseling services were available during and after data collection to assist participants who experienced emotional distress, emphasizing the study's commitment to safeguarding participant well-being.

### Inclusivity in global research practice

This study was conducted in collaboration with local researchers and community stakeholders within the refugee setting in Nigeria. Ethical, cultural, and scientific considerations were incorporated throughout the research process, including participant recruitment, informed consent, data collection, and interpretation of findings. Particular care was taken to ensure

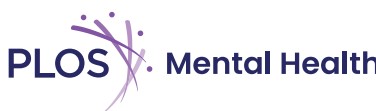

respect for the rights, dignity, and perspectives of stateless Liberian and Sierra Leonean refugee families. Additional information regarding the ethical, cultural, and scientific considerations specific to inclusivity in global research is included in the Supporting Information (S1 Checklist).

**Instruments for data collection**

The demographic variables controlled for in this study included sex, age, marital status, religion, education level, duration of stay in the refugee camp, and migration status.

Shame was assessed using Experience of Shame Scale (ESS), a 25-item measure of characterological shame (items 1–12), behavioral shame (items 13–21), and bodily shame (items 22–25). Participants responded on a four-point Likert scale ranging from 1 (Not at all) to 4 (Very much), with higher scores indicating a greater tendency to experience shame. Andrews et al. [6] demonstrated that the ESS has good validity, high internal consistency, and strong test-retest reliability. In this study, the reliability coefficient (Cronbach's alpha) was 0.93.

**Rationale for using the Experience of Shame Scale (ESS)**

Shame is often conceptualized narrowly in affective terms, which limits its applicability in understanding the full scope of psychological distress experienced by individuals, particularly in vulnerable populations like refugees. The Experience of Shame Scale (ESS) developed by Andrews et al. [6] offers a comprehensive and multidimensional approach to shame. It is one of the earliest tools to introduce domain-specific measurement of shame, addressing three distinct yet interrelated components: characterological shame (e.g., shame about who one is), behavioral shame (e.g., shame about one's actions), and bodily shame (e.g., shame related to one's body or appearance).

In the context of refugee populations, such dimensionality is essential. Refugees often grapple with complex internal experiences beyond their external hardships. Many carry deep-seated shame related to their past behaviors, identity disruptions, or physical appearance, particularly in situations involving violence, coercion, or survival-based moral compromises. The ESS is uniquely suited to uncovering these nuanced experiences that are often hidden or unspoken.

As Lemming and Boyle [7]noted, the ESS fills a significant gap in domain-specific research on shame by offering a structure that captures the internal psychological conflicts individuals may face. In our study, the ESS demonstrated excellent internal consistency (Cronbach's alpha = 0.93), supporting its use as a reliable and valid instrument for assessing shame in refugee contexts.

The Multidimensional Scale of Perceived Social Support (MSPSS) by Zimet et al. [86] is a 12-item self-report measure that assesses perceived social support (PSS) from family (items 11, 8, 4, 3), friends (items 6, 7, 9, 12), and significant others (items 1, 2, 5, 10). Participants responded on a seven-point Likert scale ranging from 1 (Very strongly disagree) to 7 (Very strongly agree), with higher scores indicating greater perceived social support. The original study reported a Cronbach's alpha of 0.90, while this study found an alpha of 0.92.

**Rationale for MSPSS selection**

Based on both existing literature and extensive experience working with refugee populations, the MSPSS was chosen for its capacity to assess perceived social support across three critical domains: family, friends, and significant others. In refugee contexts, social support networks are often fragmented or altered due to displacement, loss, or separation. Refugees may arrive with family groups requiring familial support, while others might have lost family members or been separated, relying primarily on friends or significant others. In some cases, refugees may lack both family and friend support, depending solely on significant others. The MSPSS's domain-specific design allows for a nuanced assessment of these varied support systems, reflecting the complex social realities faced by refugees. This multidimensional perspective is essential to accurately capture the role of social support in refugee mental health, making MSPSS a particularly relevant and practical choice for our study.

### The Depression, Anxiety, and Stress Scale (DASS-42)

The DASS by Lovibond and Lovibond [87] is a 42-item measure assessing depression, anxiety, and stress, with three subscales of 14 items each. Participants rated statements based on their experiences over the past week using a four-point Likert scale ranging from 0 (Did not apply to me) to 3 (Applied to me very much). Higher scores indicate greater levels of depression, anxiety, and stress, while a higher overall score reflects greater psychological distress. The DASS-42 has demonstrated high reliability, with Cronbach's alphas of 0.89 (Depression), 0.85 (Anxiety), 0.81 (Stress), and 0.95 (Total scale) [87].

### Rationale for selecting the DASS-42

The Depression, Anxiety, and Stress Scale (DASS-42) was selected for this study due to its broad and multidimensional assessment of psychological distress, as well as its applicability across diverse cultural contexts, including refugee populations. Refugees often experience cumulative psychological challenges across the migration journey—from pre-displacement anxieties about conflict and instability, through the traumatic process of displacement itself, to the post-migration stressors associated with life in refugee camps. These prolonged stressors frequently manifest in elevated levels of anxiety, chronic stress, and depressive symptoms.

The DASS-42 captures these three core domains—depression, anxiety, and stress—with sufficient depth through its 42 items (14 per subscale), allowing for a comprehensive evaluation of the psychophysiological burdens commonly observed in displaced populations. This instrument is particularly suited to refugee contexts where individuals may experience persistent fear of the unknown, social exclusion, loss of control, feelings of guilt or shame, and grief, all of which are well reflected in the symptom profiles covered by the DASS-42.

Moreover, the DASS-42 has been shown to perform well across cultures and is not bound to a specific diagnostic category, making it a flexible and culturally inclusive tool in cross-cultural mental health research. Its high internal consistency, as reported by Lovibond and Lovibond [87]—Cronbach's alphas of 0.89 (Depression), 0.85 (Anxiety), 0.81 (Stress), and 0.95 (Total scale)—further supports its psychometric robustness in diverse populations, including refugees.

### Data analysis

Descriptive statistics and Pearson's correlation analysis were conducted using IBM SPSS Version 20.0. Structural Equation Modeling (SEM) was performed in SPSS AMOS 28 using the maximum likelihood estimation method. Model fit was assessed using both absolute fit indices (e.g., chi-square [$\chi^2$] statistics) and relative fit indices, including the Standardized Root Mean Square Residual (SRMR), Root Mean Square Error of Approximation (RMSEA), and Comparative Fit Index (CFI). For absolute model fit, the $\chi^2$ value should be non-significant ($p > .05$), while for relative fit, the CFI should be greater than 0.90, and SRMR and RMSEA should be less than 0.06 [88]. The distribution of all continuous variables was moderately normal, as kurtosis and skewness values ranged between -1 and 1 [89]. Missing data were handled using **mean imputation**, as the number of missing cases was less than 5% and the data were assumed to be missing completely at random (MCAR)

### Results

Table 1 presents socio-demographic information. The sample consists mainly of males (64.2%), those of single status (54.4%), aged between18 39 (70%), Christians (90%), those who migrated involuntarily (69.6%), have stayed in the host country between 6 and 10 years (45.6%) and studied up to secondary school level (34.4%)

### Prevalence of depression, anxiety, and stress

The prevalence of depression, anxiety, and Stress is shown in Table 2 using the criteria of criteria of Lovibond and Lovibond [87]. Only about 32% had between moderate to severe depression. Approximately 40% of the sample reported severe stress levels, while more than 73% were extremely anxious.

**Table 1. Socio-demographic characteristics of participants (N = 520).**

| N = 520 Variables | n | % |
|---|---|---|
| Sex | | |
| Male | 334 | 64.2 |
| Female | 186 | 35.8 |
| Age (in years) | | |
| 14–17 | 20 | 3.8 |
| 18–28 | 189 | 36.3 |
| 29–39 | 175 | 33.7 |
| 40–50 | 85 | 16.3 |
| > 50 | 51 | 9.8 |
| Marital status | | |
| Single | 283 | 54.4 |
| Married | 171 | 32.9 |
| Divorced | 4 | .8 |
| Widowed | 62 | 11.9 |
| Religious affiliation | | |
| Christianity | 468 | 90 |
| Islam | 48 | 9.2 |
| Others | 4 | .8 |
| Education | | |
| Primary | 121 | 23.3 |
| Secondary | 179 | 34.4 |
| College | 117 | 22.5 |
| Bachelors | 93 | 17.9 |
| Postgraduate | 10 | 1.9 |
| Duration of stay | | |
| 1–5 | 100 | 19.2 |
| 6–0 | 237 | 45.6 |
| 11–15 | 92 | 17.7 |
| > 15 | 91 | 17.5 |
| Migration | | |
| Voluntary | 158 | 30.4 |
| Involuntary | 362 | 69.6 |

**Note** = number of participants; % = percentage. Age and duration of stay are reported in years. ">" indicates "greater than."

## Bivariate association among continuous variables

Table 3 displays the bivariate association among continuous variables. Experience of global shame score was negatively associated with social support global score (r=−.14, p=.001). And its dimensions of significant other support (r=−.16, p<.001), and friend support (r=−.13, p=.003)but not family support (r=−.07, p=.14). The subscales and global score of social support were significantly and negatively associated with subscales of Experience of shame (with correlation coefficients ranging between -.13 and -.19). The exceptions were the non-significant relationships of characterological shame with social support and its subscales. Also, there was no significant relationship between bodily shame and family support (r=.07, p=.13).

**Table 2. Prevalence estimates for depression, anxiety and stress.**

| DASS Rating | Depression % | Anxiety % | Stress % |
|---|---|---|---|
| Normal | 40 | 8.8 | 15.8 |
| Mild | 28.1 | 1.5 | 11.3 |
| Moderate | 28.8 | 2.7 | 31.3 |
| Severe | 3.1 | 13.7 | 33.8 |
| Extremely severe | 0 | 73.3 | 7.7 |

**Table 3. Means, standard deviations and bivariate correlations.**

| N = 520 | 1 | 2 | 3 | 4 | 5 | 6 | 7 | 8 | 9 | 10 | 11 | 13 |
|---|---|---|---|---|---|---|---|---|---|---|---|---|
| M | 30.51 | 23.06 | 9.63 | 63.20 | 18.76 | 16.80 | 16.61 | 52.17 | 11.11 | 23.55 | 23.40 | 58.05 |
| SD | 7.11 | 5.65 | 3.43 | 13.64 | 7.71 | 6.94 | 6.74 | 18.55 | 5.48 | 9.17 | 8.93 | 21.66 |
| Characterological shame (1) | – | | | | | | | | | | | |
| Behavioral shame (2) | .58** | – | | | | | | | | | | |
| Bodily shame (3) | .45** | .61** | – | | | | | | | | | |
| Experience of shame (4) | .87** | .87** | .74** | – | | | | | | | | |
| Significant other support (5) | -.06 | -.18** | -.23** | -.16** | – | | | | | | | |
| Family support (6) | .01 | -.13** | -.07 | -.07 | .60** | – | | | | | | |
| Friend support (7) | -.04 | -.18** | -.14** | -.13** | .56** | .73** | – | | | | | |
| Social support (8) | -.03 | -.19** | -.17** | -.14** | .85** | .89** | .87** | – | | | | |
| Depression (9) | .25** | .37** | .28** | .35** | -25** | -.14** | -.15** | -.21** | – | | | |
| Anxiety (10) | .28** | .38** | .39** | .40** | -.23** | -.02 | -.17** | -.16** | .74** | – | | |
| Stress (11) | .37** | .45** | .39** | .48** | -.25** | -.22** | -.25** | -.28** | .77** | .76** | – | |
| Psychological distress (12) | .33** | .44** | .40** | .46** | -.26 | -.13** | -.21** | -.24** | .89** | .93** | .93* | – |

**. Correlation is significant at the 0.01 level (2-tailed); *Correlation is significant at the 0.05 level (2-tailed)

Interpretation of Table 3

There were positive relationships between Experience of shame global scores and psychological distress (r = .46, p < .001), and its dimensions of depression (r = .35, p < .001), anxiety (r = .40, p < .001) and Stress (r = .48, p < .001). All sub-scales of Experience of shame and subscales of psychological distress were also significantly and negatively related with coefficients ranging from.25 to.45. Social support was significantly and negatively related to psychological distress, and its dimensions with correlation coefficients ranging between -.16 and -.28, p < .001. Similarly, almost all the dimensions of social support were significantly and negatively associated with all psychological distress dimensions, with correlation coefficients ranging from -.13 and -.26, p < .001. The only exception was the relationship between family support and anxiety which was not significant (r = -.02, p = .70)

This table presents descriptive statistics, including the mean (M) and standard deviation (SD), along with Pearson's correlation coefficients for key study variables among N = 520 participants. The mean values indicate the average score for each variable, while the standard deviations show the degree of variability. For instance, Characterological Shame (M = 30.51, SD = 7.11) and Behavioral Shame (M = 23.06, SD = 5.65) exhibit moderate variability, whereas Psychological Distress (M = 58.05, SD = 21.66) has a much higher SD, indicating greater variation in responses among participants.

The correlation matrix reveals significant relationships between variables. The three forms of shame—Character-ological, Behavioral, and Bodily—are highly correlated with each other, suggesting that they represent closely related

experiences. The overall Experience of Shame also shows strong positive correlations with all three types, confirming that these subscales contribute to a broader construct of shame. Notably, Behavioral Shame and Bodily Shame correlate at r=.61, while Experience of Shame correlates with Characterological Shame at r=.87, highlighting the strong interconnection between these dimensions.

Social support variables, including support from significant others, family, and friends, show negative correlations with shame and psychological distress. This suggests that higher perceived social support is associated with lower shame and distress levels. For example, Social Support (r=-.19 with Behavioral Shame) and Family Support (r=-.13 with Behavioral Shame) indicate that individuals who perceive greater support from their social network report lower levels of shame. Similarly, social support negatively correlates with Depression, Anxiety, and Stress, reinforcing its protective role in mental health.

Shame is positively correlated with psychological distress, with stronger associations observed between Experience of Shame and Anxiety (r=.40), as well as Experience of Shame and Depression (r=.35). These findings suggest that individuals who experience greater shame tend to report higher levels of emotional distress. Additionally, Stress (r=.48 with Experience of Shame) and Psychological Distress (r=.46 with Experience of Shame) further confirm that shame contributes significantly to mental health struggles.

Depression, Anxiety, and Stress are highly interrelated, indicating that these psychological distress components are not independent but rather reflect overlapping aspects of mental health difficulties. Depression and Anxiety correlate at r=.74, while Stress and Psychological Distress show an extremely high correlation (r=.93), suggesting that they measure similar emotional states. This interconnection implies that individuals experiencing one form of psychological distress are likely to experience others as well.

Overall, this table suggests that shame is a significant risk factor for psychological distress, while social support serves as a potential buffer against these negative outcomes. The findings highlight the importance of addressing shame-related experiences in mental health interventions and suggest that strengthening social support networks may help mitigate distress among affected individuals.

Fig 1 above illustrates a mediation pathway in which Experience of Shame (latent construct measured by characterological shame, behavioral shame, and bodily shame) influences Psychological Distress (latent construct measured by depression, anxiety, and stress) both directly and indirectly through Social Support (measured by perceived support from family, friends, and significant others).

Solid lines indicate standardized path coefficients. The path from Experience of Shame to Psychological Distress was significant (β=.51), as was the path from Experience of Shame to Social Support (β=−.19). The path from Social Support to Psychological Distress was not significant (β=.03, p>.05).

Covariates (age, gender, and marital status) were included in the model but had no significant effect.

**Note: ES**=Experience of Shame, **SS**=Social Support, PD=Psychological Distress. All factor loadings are significant (p<.001).

## Structural model

Fig 1 presents the model tested that tested the mediating effect of social support on the relationship between experience of shame and psychological distress, controlling for gender, age, and marital status. The factor loadings for social support and psychological distress indicators were all high and statistically significant, supporting the reliability of the measurement model.

The results indicated that experience of shame significantly predicted lower levels of social support (β=-0.19, p=.009) and higher psychological distress (β=0.51, p<.001). Although social support was significantly and negatively correlated with psychological distress at the bivariate level (r=-0.24, p<.01), its direct effect in the SEM model was non-significant (β=0.03, p>.05). This suggests that the protective role of social support may operate indirectly, possibly through reducing shame, rather than directly mitigating distress.

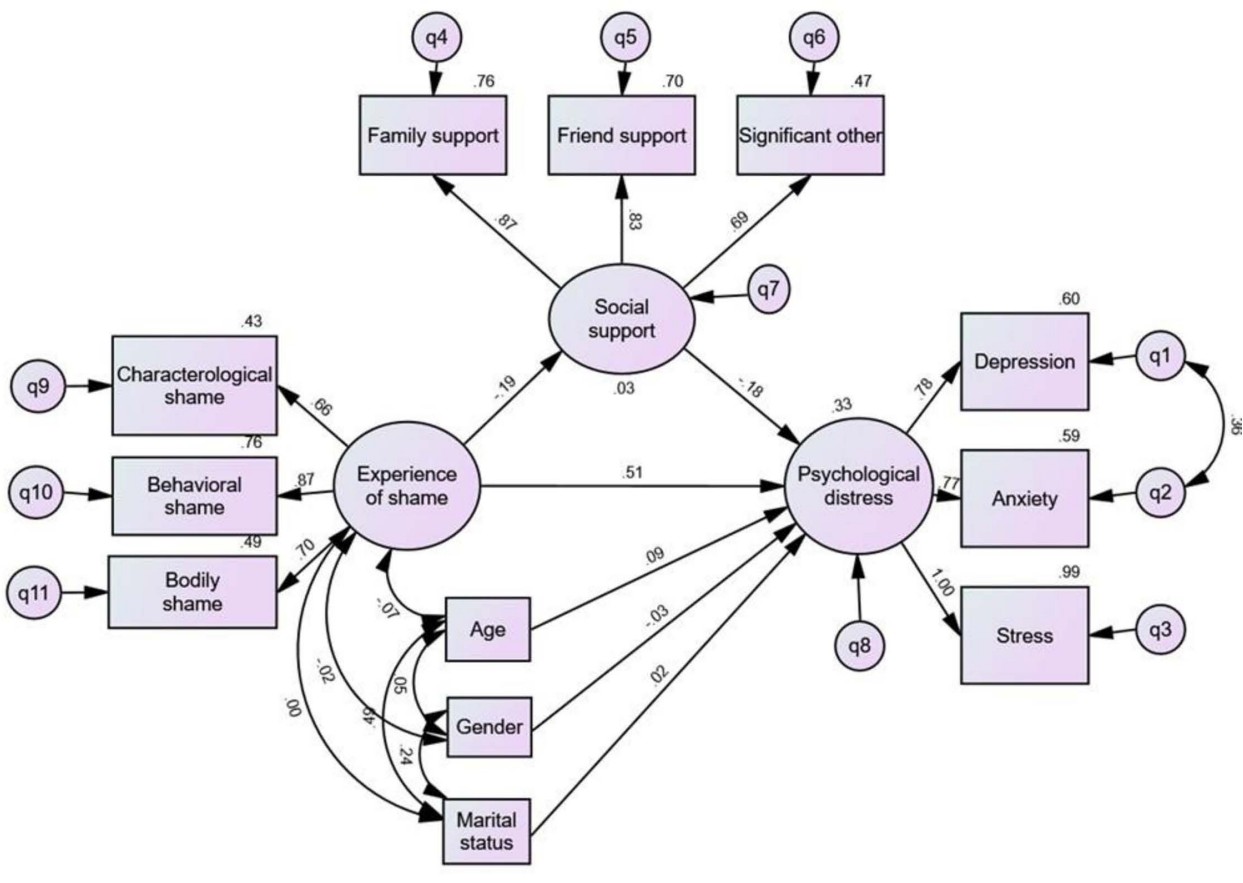

**Fig 1. Structural equation model showing the relationship between Experience of Shame, Social Support, and Psychological Distress.**

## Rationale for control variables

Gender, age, and marital status were included as control variables in this study due to their well-documented associations with both perceived social support (PSS) and psychological distress. Prior research has shown that women may report higher levels of shame and psychological distress compared to men, particularly in trauma-exposed populations [90]. Age has been shown to influence how individuals perceive and benefit from social support, particularly in emotionally demanding work environments. For instance, older nurses experienced greater psychological benefit from peer support compared to their younger counterparts, although they were less likely to receive such support during periods of distress [91]. Marital status was controlled for because being married or in a committed relationship have been consistently linked to higher levels of with perceived social support and lower levels of psychological distress in both general and refugee populations. By accounting for these demographic factors, we aimed to isolate the unique contribution of shame and PSS in predicting psychological distress among Liberian refugees.

## Model fit

The model demonstrated an adequate fit to the data: $\chi^2(44) = 192.6$, $p < .001$; RMSEA = .08 (90% CI [0.07, 0.09]); CFI = .94; SRMR = .05. The model explained 33% of the variance in psychological distress. The factor loadings for all indicators were strong and significant, providing further support for the model's measurement validity.

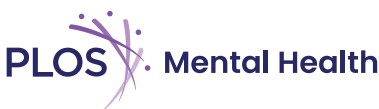

### Interpretation

These findings support a partial mediation model, where social support mediates the relationship between experience of shame and psychological distress. However, since the direct effect of shame on distress remained significant, it suggests that social support explains part—but not all—of the link between shame and distress. This implies that individuals who experience high levels of shame may suffer psychological distress not only due to reduced social support, but also because of the direct psychological impact of shame itself.

### Mediation analyses

In line with Preacher & Hayes [92], the total and indirect effects of mediation are significant. The total effect of Experience of shame on psychological distress is significant ($\beta$ = .54, p < .001). Table 3 shows the indirect effect of experience of shame on psychological distress through social support with 90% and 95% bias-corrected confidence intervals.

   Table 4 presents the bias-corrected unstandardized 90% and 95% confidence intervals for the indirect effect of experience of shame on psychological distress through social support. The total effect of experience of shame on psychological distress was statistically significant ($\beta$ = .54, p < .001), meaning that shame had a strong impact on distress before considering the mediator. The indirect effect through social support was also significant, as the confidence intervals (90% CI = .02 to .12; 95% CI = .02 to .13) did not include zero. This confirms that perceived social support mediates the relationship between shame and psychological distress. However, the direct effect of shame on distress remained significant ($\beta$ = .51, p < .001), indicating partial mediation—shame still contributes directly to psychological distress, even after accounting for the influence of social support.

### Total effect of experience of shame on psychological distress

$\beta$ = .54, p < .001 →, the total effect is statistically significant, meaning that shame has a strong impact on distress before considering the mediator.

### Indirect effect of shame on psychological distress (Through Social Support)

The confidence intervals (90% CI = .02 to .12, 95% CI = .02 to .13) do not cross zero.

   This means the indirect effect is significant, confirming that social support mediates the relationship between shame and distress.

### Partial mediation explanation

The direct effect of shame on distress remains significant ($\beta$ = .51, p < .001), even after controlling for social support.

   This indicates that while social support partially explains the relationship, shame still directly contributes to psychological distress beyond its influence on social support.

### Key findings from the Structural Model

The direct effect of experience of shame on psychological distress is strong and significant ($\beta$ = .51, p < .001), meaning that individuals with higher levels of shame experience more psychological distress.

**Table 4. Bias-corrected unstandardized 90% and 95% confidence intervals for the indirect effect of experience of shame on psychological distress through social support.**

| Mediated Path | Estimate | 90% CI Lower | 90% CI Upper | 95% CI Lower | 95% CI Upper |
|---|---|---|---|---|---|
| ES → Social Support → Psychological Distress | 0.06 | 0.02 | 0.12 | 0.02 | 0.13 |

**Note.** ES = Experience of Shame.

The direct effect of experience of shame on social support is negative and significant (β=−.19, p<.009), indicating that individuals who experience more shame perceive lower levels of social support.

The direct effect of social support on psychological distress is negative and significant (β=−.18, p<.001), meaning that individuals with higher social support report lower levels of distress.

Control variables (gender, age, marital status) do not significantly predict psychological distress, suggesting that demographic factors do not play a major role in explaining distress in this model.

## Mediation analysis

The mediation analysis follows the Preacher & Hayes [92] approach to test whether social support mediates the link between experience of shame and psychological distress.

The total effect of experience of shame on psychological distress is significant (β=.54, p<.001), indicating that shame has a strong overall impact on distress.

The indirect effect of experience of shame on psychological distress through social support is significant, as shown in Table 3.

The bias-corrected confidence intervals for the indirect effect do not pass through zero (90% CI [.02,.12]; 95% CI [.02,.13]), confirming the presence of a significant mediation effect.

Since the direct effect of experience of shame on psychological distress remains significant despite the mediation, this suggests a partial mediation. This means that while social support helps explain part of the relationship between shame and distress, shame also has a direct impact on distress beyond its effect on social support. These findings indicate that Experience of shame reduces perceived social support, and lower social support contributes to increased psychological distress. However, because shame still directly predicts distress, other mechanisms beyond social support may also play a role in this relationship.

## Interpretation of the Structural Equation Model (SEM)

This model examines the relationships between shame, social support, and psychological distress, incorporating demographic factors such as age, gender, and marital status. The standardized path coefficients quantify the strength and direction of these relationships.

The experience of shame is defined by three dimensions: characterological shame (0.43), behavioral shame (0.76), and bodily shame (0.49). Among these, behavioral shame has the strongest contribution, suggesting that shame related to one's actions plays the most significant role in the overall shame construct. Experience of shame strongly predicts psychological distress (0.51), indicating that individuals with higher levels of shame are more likely to experience increased depression, anxiety, and stress.

Social support is measured through three components: family support (0.76), friend support (0.70), and significant other support (0.47). These high factor loadings indicate that perceived support from different sources collectively forms a strong social support system. Social support is negatively correlated with Experience of shame (-0.19), suggesting that individuals with higher levels of support tend to report lower feelings of shame.

However, the indirect effect of experience of shame on psychological distress through social support was statistically significant (β=0.06; 95% CI [0.02, 0.13]), indicating a meaningful mediation pathway. However, the direct path from social support to psychological distress was small and not statistically significant (β=0.03, *p*>.05). This suggests that social support does not independently predict levels of distress in this model but instead functions as a mediator linking experience of shame to distress.

Psychological distress is represented by three indicators: depression (0.60), anxiety (0.59), and stress (0.99). Among these, stress has the highest factor loading, suggesting that it is the most dominant component of psychological distress

in this model. This indicates that individuals with higher levels of shame are particularly prone to experiencing chronic stress rather than isolated episodes of anxiety or depression.

The demographic factors—age (0.09), gender (-0.03), and marital status (0.02)—have minimal direct effects on psychological distress. However, marital status has a small negative relationship with Experience of shame (-0.24), suggesting that individuals in certain marital situations (e.g., married individuals) may report lower levels of shame compared to those who are unmarried or divorced.

The model highlights a strong association between shame and psychological distress, with social support acting as a protective factor by reducing shame rather than directly alleviating distress. Stress emerges as the most significant component of psychological distress, while demographic factors play a relatively minor role in shaping distress levels.

## Discussion

This study aimed to achieve two primary objectives: first, to examine the mediating role of perceived social support (PSS) in the relationship between shame and psychological distress, and second, to investigate the domain-specificities of PSS, shame, and psychological distress among Liberian refugees in Nigeria. To address these objectives, we formulated four hypotheses—one related to the mediating model (objective one)  and three focused on the domain-specfice relationships (objective two). Structural equation modeling (SEM) was employed to test the proposed hypotheses.f

indings from this study provide valuable insights into the mechanisms underlying refugee mental health. Regarding the first objective, the results confirmed that PSS significantly mediated the relationship between shame and psychological distress, emphasizing its role in mitigating the negative psychological effects of shame. For the second objective, the findings highlight that PSS, shame, and psychological distress interact differently across domains, suggesting that not all forms of PSS are equally effective across different types of shame or distress. This underscores the need for context-sensitive interventions tailored to refugees' specific psychosocial challenges, ultimately informing the development of targeted mental health strategies.

This mediation pathway is illustrated in the model, which depicts the structural equation model of our findings. As shown, the direct effect of experience of shame on psychological distress is strong and significant ($\beta = .51$, $p < .001$), indicating that individuals experiencing higher levels of shame report greater psychological distress. Additionally, the figure highlights the indirect pathway through which shame influences PSS ($\beta = -0.19$, $p < .001$), which in turn impacts psychological distress ($\beta = -0.18$, $p < .001$). These findings reinforce the mediating role of PSS in reducing the harmful effects of shame.

While previous research has primarily focused on PSS as a moderator among refugees, our study contributes by demonstrating the mediating role of PSS in the relationship between shame and psychological distress [13,15–18]. By uncovering this mediating mechanism, our study adds to the understanding of how social support systems operate within refugee populations, particularly in buffering against the negative impact of shame on mental well-being.

The indirect effect of experience of shame on psychological distress through social support was statistically significant ($\beta = 0.06$; 95% CI [0.02, 0.13]), indicating a meaningful mediation pathway. However, the direct path from social support to psychological distress was small and not statistically significant ($\beta = 0.03$, $p > .05$), suggesting that social support does not independently predict distress in this model.

Our findings also revealed that experience of shame was significantly and negatively associated with perceived social support, indicating that higher shame levels are linked to lower perceptions of support. While perceived social support alone did not significantly reduce distress, it served as a key mediating mechanism, meaning that shame indirectly influences psychological distress by diminishing perceived social support. This emphasizes the central role of social support in the relationship between shame and mental health. These findings suggest that strengthening social support systems in refugee contexts may help buffer the harmful emotional effects of shame, potentially reducing psychological distress in highly vulnerable populations.

These findings resonate strongly with Internal Family Systems (IFS) theory [14], which conceptualizes the human psyche as composed of multiple subparts. Shame is understood in this model as an "exiled part" — a deeply wounded internal experience that individuals push away to protect the Self. In turn, protective parts such as withdrawal, emotional detachment, or avoidance may emerge to prevent the reactivation of shame. These protectors often lead individuals to isolate themselves from social engagement. This dynamic helps explain the current finding that perceived social support was significantly reduced among those with high levels of shame, and that social support did not independently reduce psychological distress. From this perspective, shame may distort or block the internal capacity to access support — even when support is available externally.

From an IFS-informed intervention standpoint, enhancing social support alone may not be sufficient. Mental health strategies must also address internal emotional burdens by helping individuals recognize and unburden shame-based parts, restore connection with the core Self, and reduce the influence of overly protective coping strategies. Integrating emotional healing with relational support may be critical for effective trauma recovery among displaced populations.

These findings are also consistent with Complex PTSD theory (Herman, 1992; Cloitre et al., 2014), [93,94] which emphasizes that prolonged trauma often results in negative self-concept, affect dysregulation, and disturbed relationships. Shame is one of the core emotional outcomes of complex trauma and can severely damage an individual's ability to engage with others or trust available support. The observed indirect effect of shame on psychological distress through social support is in line with this theory, demonstrating that unresolved trauma may disrupt both internal regulation and external relational functioning.

In addition, the results reflect patterns described in the Social Support Deterioration Model [95], which posits that stressful or traumatic experiences can erode social support networks. In the context of stateless refugees, prolonged exposure to adversity, coupled with shame, may lead to withdrawal, mistrust, or a breakdown in social connectedness. This deterioration may be both subjective (individuals perceive less support) and objective (relationships weaken or dissolve), thus reducing support's protective value against distress.

Taken together, these theoretical perspectives suggest that interventions for refugee mental health must go beyond simply increasing social services or community support. While these are important, their effectiveness may be limited if individuals' internal emotional systems are burdened by unresolved shame. A more holistic, trauma-informed approach that addresses both the internal fragmentation caused by trauma and the external erosion of supportive relationships may be necessary for improving psychological outcomes among protracted refugee populations such as stateless Liberians.

Furthermore, our findings revealed significant direct effects, indicating that higher levels of shame were associated with lower levels of perceived social support, and conversely, higher levels of perceived social support were associated with decreased psychological distress. This suggests that shame not only directly influences psychological distress but also indirectly affects it through its impact on perceived social support. As further illustrates this indirect effect, emphasizing that PSS serves as a critical link between shame and distress. These findings suggest that enhancing social support mechanisms within refugee communities could serve as a protective factor against the detrimental effects of shame on mental health.

In summary, our study underscores the importance of perceived social support in buffering against the adverse effects of shame on psychological distress among Liberian refugees in Nigeria. By elucidating the mediating role of PSS, our findings provide valuable insights for interventions aimed at promoting mental well-being and resilience within refugee populations.

Our mediation analysis sheds light on the intricate and intertwined relationships between shame and psychological distress, echoing previous research [9,10,73] and highlighting the theoretical and practical implications of our model. Among refugees, shame often arises from psychological and emotional trauma stemming from discrimination, stigmatization, or revictimization, intertwining with a plethora of psychological symptoms including anxiety, depression, stress, loss of self-confidence, frustration, feelings of being out of control, and a desire for solitude [10,73].

Our findings suggest that refugees experiencing shame and psychological distress may find solace in the availability of social support, given its mitigating influence. High levels of perceived social support (PSS) have been shown to weaken the internalized stigma surrounding one's mental health [33], offering a pathway for individuals affected by negative stigma to cultivate positive self-awareness and alleviate negative emotions [25]. The model visually supports this theoretical argument, demonstrating that PSS acts as a critical buffer in mitigating the psychological consequences of shame.

PSS emerges as an integral aspect of a refugee's life, capable of mitigating shame and psychological distress, thus playing a crucial role in their survival. Therefore, considering PSS in the relationship between shame and psychological distress is imperative due to its potential for reducing the adverse effects of shame among refugees.

This study contributes to the existing literature by addressing PSS as a mediator between shame and psychological distress, a fact that has been overlooked previously. By closing this knowledge gap, we provide insights into the mechanisms through which social support influences the interplay between shame and psychological distress among refugees, offering valuable implications for both theory and practice in the field of refugee mental health.

In our second hypothesis, we investigated the intricate relationships between dimensions of shame and perceived social support (PSS). Our findings revealed nuanced associations, with certain dimensions of shame, such as behavioral and bodily shame, exhibiting negative correlations with specific dimensions of PSS, namely, support from friends and significant others (Behavioral shame, β = -0.19; Bodily shame, β = -0.20). However, characterological shame (β = 0.43) did not demonstrate significant correlations with any dimensions of PSS. Similarly, bodily shame showed no association with family support (β = -0.03), suggesting a lack of internal and external connectivity within these aspects of shame. While prior studies have not extensively examined how specific dimensions of shame relate to perceived social support (PSS) among refugees, the present study addresses this gap and provides empirical support for Internal Family Systems (IFS) theory [14]. IFS conceptualizes individuals as composed of multiple internal parts, including vulnerable and protective ones, whose interactions shape emotional experiences such as shame. Survival and well-being depend on maintaining core connections with family and community.

Our findings provide specific evidence for these IFS principles. Notably, bodily shame showed no significant relationship with family support (β = -0.03), suggesting that even when refugees experience shame about their physical appearance, family bonds remain a source of tangible and emotional support. This aligns with IFS's notion that protective parts work to harmonize internal conflicts and preserve essential social connections critical for survival [57].

Additionally, the differential relationships observed between characterological and behavioral shame and various dimensions of PSS reflect the nuanced dynamics of internal parts within IFS. Behavioral shame, often linked to social conformity, may facilitate adherence to culturally accepted values, helping refugees maintain social integration despite internal struggles [39]. Thus, fostering self-compassion, a core IFS therapeutic strategy, may empower individuals to externalize and improve their social relationships [58].

Culturally, the collectivist orientation among Liberian refugees likely reinforces family support as a buffering factor against psychological distress, explaining the resilience of family bonds despite shame. This culturally embedded support system exemplifies the IFS principle that core social relationships act as anchors amidst internal turmoil.

From an applied perspective, integrating self-compassion and IFS-informed approaches into refugee mental health interventions could mitigate the negative effects of shame and strengthen social support networks. Future research might also examine how demographic factors such as gender or age moderate these processes, further tailoring interventions to refugee populations.

Our third hypothesis proposed that all dimensions of perceived social support (PSS) would exhibit significant negative correlations with all dimensions of psychological distress. The results illustrated in the structural diagram strongly confirm this hypothesis, demonstrating that social support from family, friends, and significant others is inversely related to depression, anxiety, and stress among Liberian refugees in Nigeria.

As shown in the diagram, the standardized path coefficients highlight the magnitude of these relationships. Notably, social support negatively predicted psychological distress (β = -0.18), reinforcing the buffering effect of PSS against distress. Furthermore, social support had direct paths to its three subcomponents—family support (β = 0.87), friend support (β = 0.83), and significant other support (β = 0.69), while psychological distress was composed of depression (β = 0.60), anxiety (β = 0.59), and stress (β = 0.99). These findings suggest that reductions in social support are linked to heightened distress levels, aligning with prior studies in general populations [65,67,69].

This structural model also underscores the mediating role of social support in mental health outcomes among refugees. Given the significant pathways observed, our results emphasize the importance of strengthening social bonds to mitigate psychological distress. Research has consistently demonstrated that perceived social support enhances psychological resilience by fostering emotional regulation and providing coping strategies [96,97]. Additionally, a strong sense of belonging—enhanced through family cohesion, friendships, and significant others—has been linked to lower distress levels and better mental health outcomes [20,54].

In sum, the model highlights the critical role of perceived social support in reducing psychological distress, reinforcing the need for targeted psychosocial interventions. Strengthening refugee social networks, facilitating support systems, and fostering interpersonal connections are essential for promoting resilience and improving well-being among refugee populations. These findings contribute to a deeper understanding of how self-conscious emotions, such as shame, can influence psychological distress through perceived social support. While the current study employed a mediation model, it is important to acknowledge that more complex frameworks—such as moderated mediation—may better capture the interplay between emotional experiences and social resources. The strength of these indirect effects may vary depending on factors such as gender, trauma severity, or cultural background. This possibility is explored further in the Limitations section and represents a valuable direction for future research.

Our final hypothesis further confirmed the significant relationships between all dimensions of shame and psychological distress. Consistent with prior research, our findings highlight the high prevalence of psychological distress among Liberian refugees in Nigeria, with approximately 76.4% experiencing PTSD, often comorbid with depression, anxiety, and stress [3]

. Notably, the diagram visually demonstrates these relationships, emphasizing that shame-related experiences are strongly associated with heightened psychological distress.

As illustrated in the structural model, the standardized path coefficients indicate that characterological shame (β = 0.74), bodily shame (β = 0.65), and behavioral shame (β = 0.71) each significantly predict increased psychological distress. These findings align with the growing recognition of shame as a key factor in PTSD symptomatology, classified under persistent negative emotional states in the DSM-5 [12]. Shame, as a deeply distressing self-conscious emotion, is intricately tied to self-perception, social worth, and identity, often resulting in avoidance and withdrawal behaviors [21].

The results in the visual presentation further reveal that each dimension of shame uniquely correlates with different aspects of psychological distress. Specifically, characterological shame exhibited the strongest associations with anxiety (β = 0.79) and depression (β = 0.76), suggesting that shame rooted in self-perception may fuel internalized distress and intra-psychic conflict [80]. Similarly, behavioral shame (β = 0.71) significantly predicted stress levels, supporting previous findings that shame linked to one's actions may contribute to maladaptive coping and emotional dysregulation [80].

Moreover, our study reinforces the domain-specificity of shame, as proposed by Andrews et al. [6]. The diagram illustrates how bodily shame uniquely predicts increased anxiety (β = 0.65), further emphasizing how appearance-related self-conscious emotions influence psychological distress [97]. These results support prior studies emphasizing that shame manifests differently across cognitive, emotional, and social domains, impacting mental health in distinct ways [80,81].

Interestingly, while our findings align with previous research linking shame to distress [80], some studies [6,36,98] have suggested that behavioral shame may be associated with future-oriented behavior and control-oriented attributions [99]. These discrepancies could be attributed to differences in study populations, with prior research focusing on general

populations, while our study examines a clinical sample of Liberian refugees who have experienced prolonged displacement and trauma for over 30 years [45].

Overall, the result substantiates the critical role of shame as a significant psychological stressor among refugees, reinforcing the need for trauma-informed interventions that address shame-related distress. Future research should further explore culturally sensitive therapeutic approaches aimed at reducing self-stigmatization and emotional withdrawal, thereby mitigating the adverse psychological effects of shame in refugee populations.

## Implications of the study

This study advances understanding of shame and psychological distress among Liberian refugees by demonstrating domain-specific effects of different shame types on mental health outcomes. These findings align with the Internal Family Systems (IFS) framework, which conceptualizes individuals as comprising multiple internal parts, some burdened by shame and others embodying a core, compassionate self. Specifically, the finding that bodily shame showed no significant impact on family support resonates with IFS's principle that certain internal parts may be protected or harmonized within the individual's self-system, allowing social support to remain intact despite internal distress. This supports the idea that interventions enhancing self-compassion and integration of internal parts, as proposed in IFS therapy, may be effective in reducing shame-related distress and improving social connections.

Clinically, these insights highlight the value of trauma-informed and culturally sensitive therapies that address shame's complexity. IFS therapy's focus on integrating dissociated self-aspects and cultivating a compassionate self offers a promising approach to reconcile intrapsychic conflicts reflected in our findings. Combined with Compassion-Focused Therapy (CFT) and trauma-focused interventions, such approaches could be tailored to address specific shame domains, ultimately strengthening perceived social support and mitigating psychological distress among refugees.

Future research should further explore the mechanisms by which IFS-informed interventions impact the nuanced dimensions of shame and social support in refugee populations, supporting development of targeted mental health strategies sensitive to the complex internal and external dynamics revealed in this study.

Given the deeply interpersonal nature of shame, culturally adapted group therapy that fosters communal healing and peer support could provide significant benefits. Encouraging self-expression through storytelling, shared experiences, and traditional cultural rituals may help refugees reframe their experiences of shame and enhance their sense of social belonging. Since shame often reinforces social withdrawal, group-based interventions that provide safe spaces for refugees to share their narratives can mitigate the isolating effects of shame and promote psychological healing.

Beyond clinical applications, our findings highlight the urgent need for systemic interventions to address the structural and social determinants of shame and psychological distress among refugees. Policymakers, humanitarian agencies, and mental health practitioners should collaborate to develop comprehensive psychosocial support programs that target shame-related distress. Mental health policies should recognize shame as a core component of refugee distress and integrate psychoeducation on shame resilience into resettlement and support services. Long-term mental health support must be prioritized in post-migration settings, where shame and psychological distress may intensify due to discrimination, poverty, and social exclusion.

Additionally, programs that promote social reintegration, such as vocational training, peer mentorship, and community-based support groups, could help mitigate socially induced shame and restore a sense of dignity among refugees. Training for humanitarian workers and mental health practitioners should include specialized education on shame-informed care, equipping them with the skills to recognize and address shame-related distress effectively. Integrating mental health literacy into refugee assistance programs may also reduce stigma, encourage help-seeking behaviors, and normalize discussions around psychological distress.

Addressing the systemic causes of refugee distress, such as statelessness, discrimination, and barriers to education and employment, is equally critical. Advocacy efforts should focus on securing legal recognition and integration programs

for refugees, as this can alleviate institutionalized forms of shame and reinforce a sense of agency and self-worth. Providing pathways to citizenship, work opportunities, and social inclusion initiatives could significantly reduce the psychological burden associated with displacement and marginalization.

By embracing a multifaceted approach that integrates clinical, social, and policy-level interventions, we can foster greater resilience and psychological well-being among refugees. Future studies should examine longitudinal trajectories of shame and distress, considering how post-migration experiences shape mental health outcomes. Moreover, culturally tailored interventions that address shame, self-worth, and trauma recovery should be prioritized in refugee mental health programs.

In addition, the finding that perceived social support mediates the relationship between shame and psychological distress emphasized the importance of targeting emotional barriers when designing support systems for refugees. Future research could build on this by examining whether the effectiveness of perceived social support varies across subgroups, such as by gender, age, or trauma severity. Identifying such moderators would not only advance theoretical understanding but also help tailor psychosocial interventions more precisely to those most in need.

Moreover, addressing shame as both a psychological and social phenomenon is essential for promoting healing, empowerment, and successful integration within refugee populations. Continued research and advocacy are necessary to ensure that mental health services are accessible, trauma-informed, and culturally appropriate—ultimately mitigating the long-term psychological consequences of forced displacement.

## Limitations of the study

While the present study provides valuable insights, several limitations must be acknowledged. First, data were collected exclusively from a rural refugee camp, which may limit the generalizability of findings to refugee populations in urban settings or those integrated into host communities. Future research should include urban refugee populations to examine potential differences in the role of perceived social support (PSS) in mediating the relationship between shame and psychological distress. Such comparative analyses would contribute to a broader understanding of these dynamics across diverse living conditions.

Second, the study employed a cross-sectional design, which restricts the ability to establish causal relationships among shame, perceived social support, and psychological distress. Longitudinal studies would be beneficial in examining how these variables interact and evolve over time.

Additionally, while this study conceptualized PSS as a mediator between shame and psychological distress, future research could explore PSS as a moderating variable to determine whether it buffers the psychological impact of shame. Investigating this buffering effect could yield critical insights into the protective role of social support, thereby informing interventions aimed at strengthening support networks in refugee communities.

Furthermore, the transferability of our findings to other host countries and refugee groups remains uncertain due to contextual differences, including geopolitical factors, legal frameworks, and resource availability. Conducting cross-cultural and longitudinal studies across diverse refugee populations would help assess the stability of these findings and support the development of context-specific intervention strategies.

Moreover, while the current study focused on a mediation model examining the indirect effects of perceived social support and shame on psychological distress and its dimensions among Liberian refugees in Nigeria, it is important to acknowledge that alternative model specifications may also be plausible. For example, a moderated mediation model could reveal that the strength or direction of the mediation effects varies based on factors such as gender, age, or length of displacement. Although such alternatives were beyond the scope of this analysis, future research could explore these models to offer a more nuanced understanding of how social and emotional resources interact with refugee experiences to influence mental health outcomes.

More so, given the closure of the refugee camp and the revocation of refugee status for Liberian refugees in Nigeria, future research should examine the long-term consequences of these policy shifts. Investigating how displaced individuals navigate post-camp life without formal support structures — including their adaptive strategies and resilience mechanisms — could offer valuable insights for policymakers and humanitarian organizations aiming to enhance refugee integration and well-being in similar protracted displacement contexts.

Lastly, another limitation concerns the reliance on self-report measures for assessing constructs such as shame and psychological distress. Given the sensitivity of these experiences, particularly within post-conflict refugee settings, responses may be influenced by social desirability bias, emotional avoidance, or fear of stigma. This could lead to underreporting or distorted responses, potentially affecting the accuracy of the data. Future research may benefit from incorporating clinician-administered interviews or mixed-methods approaches to capture these experiences more comprehensively.

## Conclusion

Having resided in Nigeria for over three decades, Liberian refugees continue to experience significant psychological distress. This study examined the mediating role of perceived social support (PSS) in the relationship between shame and psychological distress and explored domain-specific patterns among these constructs. Our findings confirm that PSS significantly mediates the effect of shame on distress, reinforcing its protective function in refugee mental health.

From an Internal Family Systems (IFS) perspective, these results underscore the importance of restoring relational connections to alleviate the burden of shame carried by internal parts. The observed domain-specific variations—where behavioral and bodily shame showed stronger associations with PSS, but characterological shame did not—reflect how some shame-related internal parts are more accessible to external support, while others may remain isolated or highly protected.

Strengthening social connections, particularly within family and peer networks, aligns with IFS principles of healing through connection and internal harmony. Given the post-camp challenges and prolonged displacement, future research should explore how domain-specific PSS sustains resilience over time. Early recognition and intervention focused on integrating shame-related parts and enhancing support systems are essential for fostering long-term mental health and social integration in displaced populations.

## Supporting information

**S1 Checklist. Inclusivity in Global Research Statement.**
(DOCX)

**S1 File. Questionnaire Used for Data Collection.**
(PDF)

## Author contributions

**Conceptualization:** Dogbahgen Alphonso Yarseah.

**Data curation:** Ololade Omolayo Ogunsanmi, Olu Francis Ibimiluyi, Elijah Olawale Olaoye, Esu Stanley Ezeani, Viola H. Cheeseman.

**Formal analysis:** Ololade Omolayo Ogunsanmi, Joyce Olufunke Ogunsanmi, Olu Francis Ibimiluyi, Viola H. Cheeseman.

**Funding acquisition:** Olu Francis Ibimiluyi.

**Investigation:** Joyce Olufunke Ogunsanmi, Elijah Olawale Olaoye, Esu Stanley Ezeani.

**Methodology:** Dogbahgen Alphonso Yarseah, Ololade Omolayo Ogunsanmi, Joyce Olufunke Ogunsanmi, Elijah Olawale Olaoye, Viola H. Cheeseman.

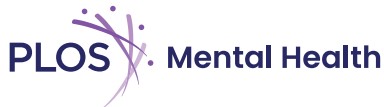

**Project administration:** Esu Stanley Ezeani.

**Resources:** Esu Stanley Ezeani.

**Software:** Viola H. Cheeseman.

**Validation:** Elijah Olawale Olaoye, Viola H. Cheeseman.

**Visualization:** Joyce Olufunke Ogunsanmi, Elijah Olawale Olaoye, Viola H. Cheeseman.

**Writing – original draft:** Dogbahgen Alphonso Yarseah, Ololade Omolayo Ogunsanmi.

**Writing – review & editing:** Dogbahgen Alphonso Yarseah, Joyce Olufunke Ogunsanmi.

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
