## [Decision Letter · Decision Letter 0]

12 May 2025

PMEN-D-25-00158

The Mediating Effects of Perceived Social Support and Shame on Psychological Distress and Its Dimensions among Liberian Refugees in Nigeria

PLOS Mental Health

Dear Dr. Yarseah,

Thank you for submitting your manuscript to PLOS Mental Health. After careful consideration, we feel that it has merit but does not fully meet PLOS Mental Health’s publication criteria as it currently stands. Therefore, we invite you to submit a revised version of the manuscript that addresses the points raised during the review process.

We look forward to receiving your revised manuscript.

Kind regards,

David Onchonga, Ph.D.

Academic Editor

PLOS Mental Health

Journal Requirements:

1. Please include a complete copy of PLOS’ questionnaire on inclusivity in global research in your revised manuscript. Our policy for research in this area aims to improve transparency in the reporting of research performed outside of researchers’ own country or community. The policy applies to researchers who have travelled to a different country to conduct research, research with Indigenous populations or their lands, and research on cultural artefacts. The questionnaire can also be requested at the journal’s discretion for any other submissions, even if these conditions are not met.  Please find more information on the policy and a link to download a blank copy of the questionnaire here: https://journals.plos.org/mentalhealth/s/best-practices-in-research-reporting. Please upload a completed version of your questionnaire as Supporting Information when you resubmit your manuscript. 2. In the online submission form, you indicated that Data will be available through reasonable request to the corresponding author. All PLOS journals now require all data underlying the findings described in their manuscript to be freely available to other researchers, either 1. In a public repository, 2. Within the manuscript itself, or 3. Uploaded as supplementary information. This policy applies to all data except where public deposition would breach compliance with the protocol approved by your research ethics board. If your data cannot be made publicly available for ethical or legal reasons (e.g., public availability would compromise patient privacy), please explain your reasons by return email and your exemption request will be escalated to the editor for approval. Your exemption request will be handled independently and will not hold up the peer review process, but will need to be resolved should your manuscript be accepted for publication. One of the Editorial team will then be in touch if there are any issues. 3. We have amended your Competing Interest statement to comply with journal style. We kindly ask that you double check the statement and let us know if anything is incorrect. 4. Please provide separate figure files in .tif or .eps format. For more information about figure files please see our guidelines:  https://journals.plos.org/mentalhealth/s/figures https://journals.plos.org/mentalhealth/s/figures#loc-file-requirements 5. Please upload a copy of Figure 2 which you refer to in your text on page 49, 53. Or, if the figure is no longer to be included as part of the submission please remove all reference to it within the text.

Additional Editor Comments (if provided):

Reviewers' comments:

Reviewer's Responses to Questions

**Comments to the Author**

1. Does this manuscript meet PLOS Mental Health’s publication criteria? Is the manuscript technically sound, and do the data support the conclusions? The manuscript must describe methodologically and ethically rigorous research with conclusions that are appropriately drawn based on the data presented.

Reviewer #1: Partly

Reviewer #2: No

2. Has the statistical analysis been performed appropriately and rigorously?

Reviewer #1: Yes

Reviewer #2: Yes

3. Have the authors made all data underlying the findings in their manuscript fully available (please refer to the Data Availability Statement at the start of the manuscript PDF file)?

Reviewer #1: No

Reviewer #2: Yes

4. Is the manuscript presented in an intelligible fashion and written in standard English?

Reviewer #1: Yes

Reviewer #2: Yes

5. Review Comments to the Author

Reviewer #1: This is a timely and relevant study examining the mediating effects of perceived social support (PSS) and shame on psychological distress among Liberian refugees in Nigeria. The study addresses a clear gap in the literature, and the hypotheses are well stated. The methodology is appropriate, and the conclusions are generally supported by the findings. However, the introduction is repetitive in places and would benefit from a more concise and focused presentation. The section discussing Internal Family Systems (IFS) therapy needs clearer integration with the study’s findings. Additionally, the manuscript should include a brief limitations section to address issues such as its cross-sectional design, use of self-report data, and generalisability.

Statistical Analysis

The statistical analysis is appropriate and robust. The use of validated instruments (ESS, MSPSS, and DASS-42) and structural equation modelling (SEM) is well justified. The mediation analysis is well executed, using established methods and confidence intervals.

To improve this:

Provide a rationale for including gender, age, and marital status as control variables.

Consider reporting effect sizes alongside beta values.

Confirm whether assumptions for SEM (such as multivariate normality) were tested and met.

Data Availability

The current data availability statement does not comply with PLOS’s open data policy. Stating that data are "available on reasonable request" is insufficient. The authors should deposit the dataset in a publicly accessible repository and update the statement accordingly.

Presentation and Use of English

The manuscript is generally intelligible but requires moderate editing for grammar and clarity. The introduction includes repeated content and could be shortened.

Some examples of awkward or incorrect phrasing include:

“The will be a significant relationship…” (should read “There will be…”)

“Domain-specificity of how specific sources…” (unclear wording)

Overuse of certain references, such as Yarseah (2016) and Womersley (2022), which appear multiple times.

The authors should revise the text for grammar, syntax, and clarity. As PLOS does not provide copyediting, the language must be clear and accurate before publication.

Reviewer #2: This study is significant, precise and reflects both the psychological depth and humanitarian relevance of the study. It promises a meaningful investigation into complex emotional and social dynamics within a vulnerable population.

1. Condense the introduction to focus on the most essential information: the context of displacement, the importance of the study, and the gaps in existing research. Leave out detailed background on Liberia's civil war or the global refugee statistics unless they are directly relevant to the research questions.

2. While the psychometrics of the scales are described, a brief rationale for why each was chosen in the context of refugee populations would strengthen the justification of your methods. The phrase "leading replacement method" is ambiguous. It would help to clarify whether this refers to mean imputation, last observation carried forward, or another specific technique.

3. The study does not mention social desirability bias or interviewer influence, which are possible concerns when verbal assistance is involved. A brief note on steps taken to minimize such bias would enhance transparency.

4. The variables should be clearly labeled in the table. Use consistent variable names throughout. Sometimes you refer to “Experience of Shame,” other times just “Shame” or “ES. The text says “social support on psychological distress is weak (0.03, non-significant)”—yet earlier it is reported as significant (β = -0.18, p < .001). This inconsistency needs correction or clarification. Your SEM results are rich, but there's no mention of model fit indices (e.g., RMSEA, CFI, TLI, SRMR). Without these, it’s difficult to assess how well the model fits the data. These should be included if available.

5. The term “Figure 1” is repeated too frequently in close succession, which creates redundancy. Once the figure has been introduced and described, it’s best to avoid continually referencing it with “as shown in Figure 1” or “Figure 1 illustrates…” unless you’re presenting new data from it.

6. While well-referenced, certain citations (e.g., Schwartz, 2013; 2020) are repeated multiple times in rapid succession. Group or combine such references where appropriate to avoid the appearance of overcitation.

7. Briefly discussing ethical dimensions (e.g., informed consent, trauma sensitivity) could strengthen the study’s research ethics positioning.

8. Terms such as "characterological shame" and "bodily shame" may not be familiar to all readers. A brief clarification or more context around these terms would be helpful, especially since this is a study focusing on refugees who may not have the same academic background as typical participants in psychological research.

6. PLOS authors have the option to publish the peer review history of their article (what does this mean?). If published, this will include your full peer review and any attached files.

**Do you want your identity to be public for this peer review?** For information about this choice, including consent withdrawal, please see our Privacy Policy.

Reviewer #1: **Yes: **Simon Browes

Reviewer #2: **Yes: **Riyadh Hossain

---

## [Decision Letter · Decision Letter 1]

13 Jun 2025

PMEN-D-25-00158R1

The Mediating Effects of Perceived Social Support and Shame on Psychological Distress and Its Dimensions among Liberian Refugees in Nigeria

PLOS Mental Health

Dear Dr. Yarseah,

Thank you for submitting your manuscript to PLOS Mental Health. After careful consideration, we feel that it has merit but does not fully meet PLOS Mental Health’s publication criteria as it currently stands. Therefore, we invite you to submit a revised version of the manuscript that addresses the points raised during the review process.

We look forward to receiving your revised manuscript.

Kind regards,

David Onchonga, Ph.D.

Academic Editor

PLOS Mental Health

Journal Requirements:

Reviewers' comments:

Reviewer's Responses to Questions

**Comments to the Author**

1. If the authors have adequately addressed your comments raised in a previous round of review and you feel that this manuscript is now acceptable for publication, you may indicate that here to bypass the “Comments to the Author” section, enter your conflict of interest statement in the “Confidential to Editor” section, and submit your "Accept" recommendation.

Reviewer #1: All comments have been addressed

Reviewer #2: All comments have been addressed

2. Does this manuscript meet PLOS Mental Health’s publication criteria? Is the manuscript technically sound, and do the data support the conclusions? The manuscript must describe methodologically and ethically rigorous research with conclusions that are appropriately drawn based on the data presented.

Reviewer #1: Yes

Reviewer #2: Yes

3. Has the statistical analysis been performed appropriately and rigorously?

Reviewer #1: Yes

Reviewer #2: Yes

4. Have the authors made all data underlying the findings in their manuscript fully available (please refer to the Data Availability Statement at the start of the manuscript PDF file)?

Reviewer #1: Yes

Reviewer #2: Yes

5. Is the manuscript presented in an intelligible fashion and written in standard English?

Reviewer #1: No

Reviewer #2: Yes

6. Review Comments to the Author

Reviewer #1: Thank you for your resubmission. This is a timely and methodologically sound study. The manuscript addresses an important gap in refugee mental health research and offers insight into psychological processes that may inform culturally sensitive interventions.

Strengths:

- The study is ethically and methodologically rigorous.

- Use of validated instruments with high internal reliability is appropriate.

- Structural Equation Modeling is suitably applied, with well-reported fit indices.

- Conceptual framing using Internal Family Systems (IFS) theory adds interpretive depth.

- Conclusions are well-supported by the data.

Suggestions for Improvement:

1. Language and Presentation:

- The manuscript would benefit from thorough copyediting to address typographical and grammatical errors.

- Several phrases are awkward or imprecise (e.g., "Table 3 displaced" should be "displays").

- There are copy-paste artifacts that may cause confusion (e.g., a reference to “suicidal PTSD” in Table 4 that is not applicable to this study).

2. Statistical Analysis:

- The analysis is robust, but further clarity around the non-significant direct path from social support to distress would strengthen interpretation.

- Consider acknowledging the possibility of alternative model specifications (e.g., moderated mediation), as you briefly mention in your discussion.

3. Figures and Tables:

- Ensure that all figures (particularly Figure 1 and Figure 2) are present, legible, and clearly described. Currently, figure placement and description are insufficient for full appraisal.

4. Limitations:

- Consider reinforcing the limitations related to self-report bias and the potential sensitivity of constructs like shame and distress in post-conflict refugee contexts.

Overall, the study offers a compelling and important contribution to the literature. Addressing the points above will enhance its clarity and impact.

Reviewer #2: (No Response)

7. PLOS authors have the option to publish the peer review history of their article (what does this mean?). If published, this will include your full peer review and any attached files.

**Do you want your identity to be public for this peer review?** For information about this choice, including consent withdrawal, please see our Privacy Policy.

Reviewer #1: **Yes: **Simon Browes

Reviewer #2: **Yes: **Riyadh Hossain

---

## [Editor Report · Decision Letter 2]

24 Jun 2025

PMEN-D-25-00158R2

The Mediating Effects of Perceived Social Support and Shame on Psychological Distress and Its Dimensions among Liberian Refugees in Nigeria

PLOS Mental Health

Dear Dr. Yarseah,

Thank you for submitting your manuscript to PLOS Mental Health. After careful consideration, we feel that it has merit but does not fully meet PLOS Mental Health’s publication criteria as it currently stands. Therefore, we invite you to submit a revised version of the manuscript that addresses the points raised during the review process.

We look forward to receiving your revised manuscript.

Kind regards,

David Onchonga, Ph.D.

Academic Editor

PLOS Mental Health
---

## [Editor Report · Decision Letter 3]

9 Jul 2025

The Mediating Effects of Perceived Social Support and Shame on Psychological Distress and Its Dimensions among Liberian Refugees in Nigeria

PMEN-D-25-00158R3

Dear Mr. Yarseah,

We are pleased to inform you that your manuscript 'The Mediating Effects of Perceived Social Support and Shame on Psychological Distress and Its Dimensions among Liberian Refugees in Nigeria' has been provisionally accepted for publication in PLOS Mental Health.

Best regards,

David Onchonga, Ph.D.

Academic Editor

PLOS Mental Health